# 360-MLC: Multi-view Layout Consistency for Self-training and Hyper-parameter Tuning

**Bolivar Solarte**[*1]**, Chin-Hsuan Wu**[*1]**, Yueh-Cheng Liu**[1]**, Yi-Hsuan Tsai**[†2]**, Min Sun**[1]
[1]National Tsing Hua University, [2]Phiar Technologies

https://enriquesolarte.github.io/360-mlc

## Abstract

We present 360-MLC, a self-training method based on multi-view layout consistency for finetuning monocular room-layout models using unlabeled 360-images only. This can be valuable in practical scenarios where a pre-trained model needs to be adapted to a new data domain without using any ground truth annotations. Our simple yet effective assumption is that multiple layout estimations in the same scene must define a consistent geometry regardless of their camera positions. Based on this idea, we leverage a pre-trained model to project estimated layout boundaries from several camera views into the 3D world coordinate. Then, we re-project them back to the spherical coordinate and build a probability function, from which we sample the pseudo-labels for self-training. To handle unconfident pseudo-labels, we evaluate the variance in the re-projected boundaries as an uncertainty value to weight each pseudo-label in our loss function during training. In addition, since ground truth annotations are not available during training nor in testing, we leverage the entropy information in multiple layout estimations as a quantitative metric to measure the geometry consistency of the scene, allowing us to evaluate any layout estimator for hyper-parameter tuning, including model selection without ground truth annotations. Experimental results show that our solution achieves favorable performance against state-of-the-art methods when self-training from three publicly available source datasets to a unique, newly labeled dataset consisting of multi-view images of the same scenes.

## 1   Introduction

Room-layout geometry is one of the fundamental geometry representations for an indoor scene, which can be parameterized with points and lines describing corners and wall boundaries. Therefore, this geometry has been largely used as a primary stage for challenging tasks like robot localization [5, 42], scene understanding [24], floor plan estimation [29], etc. Several methods have been proposed for estimating layout geometry from imagery [12, 13, 45], while the current state-of-the-art methods [19, 10, 17, 47, 15] leverage deep learning approaches to regress the wall-ceiling and floor boundaries directly from monocular 360-images in a supervised manner.

However, deploying a room-layout model using 360-images in a new target domain remains a challenging problem. For instance, a pre-trained layout model may estimate inconsistent geometry, due to novel view positions, different lighting conditions, or severe object occlusions in the scene, especially for large and complex rooms. To handle these issues, ground truth annotations in the new target domain are usually required to finetune the model, which involves a cumbersome data labeling process. Moreover, considering the large variety of indoor scene styles, a room-layout model may

---

[*]The authors contribute equally to this paper.
[†]Currently at Google.

36th Conference on Neural Information Processing Systems (NeurIPS 2022).

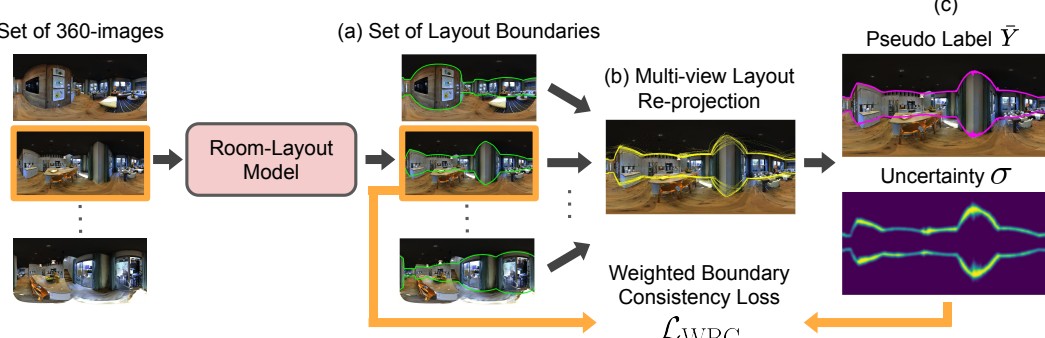

Figure 1: **360-MLC pipeline.** (a) Our method uses a pre-trained room-layout model and collects the layout estimations from multiple views in the same scenes; (b) we re-project multi-view estimates to the target view (the image with the orange bounding box); (c) 360-MLC then generates the pseudo-label and the uncertainty to compute the consistency loss as self-supervisions without requiring ground truth labels in the target scene.

demand more new labeled data to adapt to another target domain. In this paper, we aim to make a pre-trained room-layout model self-trainable in the new target domain without using any annotated data during model finetuning, in which such setting is practical but has not been studied widely for room-layout estimation.

To this end, we present 360-MLC, a self-training method based on Multi-view Layout Consistency (MLC), capable of adapting a pre-trained layout model into a new data domain using solely registered 360-images. Our main assumption is that multiple layout estimates must define together a consistent scene geometry regardless of their camera positions. Based on this idea, we project several estimated layout boundaries into the world coordinate with the Euclidean space and re-project them back to a target view (spherical coordinate) for building a probability function, from where we sample the most likely points as pseudo-labels. In addition, we evaluate the variance in the re-projected boundaries as an uncertainty measure to weight our pseudo-labels for unconfident boundary regions. We combine both our pseudo-label and its uncertainty into the proposed *weighted boundary consistency loss* ($\mathcal{L}_{\text{WBC}}$), which allows us to finetune a layout model in a self-training manner without requiring any ground truth annotations. Therefore, pseudo-labels generated via our MLC have higher quality than pseudo-labels predicted from a single view, which is crucial for achieving satisfactory performance when no annotations are available in the new domain.

We further propose the *multi-view layout consistency metric* based on entropy information ($\mathcal{H}_{\text{MLC}}$) that quantitatively evaluates any layout predictor without requiring ground truth labels. This metric can be used to monitor the quality of the predicted layouts in both training and testing. As a result, our $\mathcal{H}_{\text{MLC}}$ metric can be used for hyper-parameter tuning and model selection, which is practical when no ground truths are available in the new domain. Our key idea behind $\mathcal{H}_{\text{MLC}}$ is to evaluate the entropy information from multiple layouts projected into a discrete grid as a 2D density function, where a better geometry alignment of layouts would yield a lower entropy evaluation. We leverage our $\mathcal{H}_{\text{MLC}}$ metric across different experiments and show its versatility and reliability for the multi-view layout setting.

In experiments, we leverage the MP3D-FPE [29] multi-view dataset as our unlabeled new domain. Since layout ground truths are not available in this dataset, we manually annotate 7 scenes with 700 frames for only the performance evaluation purpose. For the dataset used in model pre-training, we validate our framework on three real-world room-layout estimation datasets: MatterportLayout [36, 48], Zillow Indoor Dataset (ZInD) [9], and the dataset used in LayoutNet [47]. We show that our method is able to handle two practical settings during training: 1) having both the labeled pre-trained data and the unlabeled data in the target domain, 2) only providing the unlabeled data with the pre-trained model. Moreover, we demonstrate the usefulness of the proposed modules, including the weighted boundary consistency loss with pseudo-labels, and the multi-view layout consistency metric that facilitates the hyper-parameter tuning without the need for ground truth labels.

Our contributions based on the idea of Multi-view Layout Consistency are summarized as follows:

1. Based on multi-view geometry consistency, we propose a self-training framework for room-layout estimation, requiring only registered 360-images as the input.

2. We propose the *weighted boundary consistency loss function* ($\mathcal{L}_{\text{WBC}}$) that uses multiple estimated layout boundaries with their uncertainty to improve the self-training process using pseudo-labels.

3. We introduce the *multi-view layout consistency metric* ($\mathcal{H}_{\text{MLC}}$) for measuring multi-view layout geometry consistency based on entropy information. This metric allows us to evaluate any layout estimator quantitatively without requiring ground truth labels, which enables hyper-parameter tuning and model selection.

## 2 Related Work

**Indoor room layout estimation.** Estimating the layout structure of a room for cluttered indoor environments is a challenging task. Early methods estimate plane surfaces and their orientations based on points or edge features to construct the spatial layouts for perspective images [12, 13, 35] or panorama images (i.e. 360-images) [45, 41]. On the other hand, some approaches [3, 14, 7, 40] leverage semantic cues in the scene, such as objects or humans, to improve the layout estimation.

Deep learning approaches [19, 10, 17, 47, 15] leverage convolutional neural networks (CNN) to extract the geometric cues (e.g., corners or edges) and semantic cues (e.g., pixel-wise segmentation), which largely enhance the performance. Recent state-of-the-art methods [43, 30, 31, 37] are able to robustly estimate the layout of the whole room from a single panorama image. CFL [11] and AtlantaNet [21] avoid the commonly-used Manhattan world assumption [8], enabling the ability to handle complex room shapes. In addition to monocular approaches, MVLayoutNet [16] and PSMNet [38] explore the usage of multi-view 360-images as inputs to further improve the layout estimations. However, training deep neural networks to perform layout estimation requires large-scale datasets with manual annotations. In this paper, we aim to mitigate this problem by learning from unlabeled data.

**Self-training.** To incorporate unlabeled data during training, self-training [27, 44, 23] uses a pre-trained teacher model to generate pseudo-labels for the data without ground truth annotations. Then, both the labeled data and the unlabeled data with pseudo-labels are used to train a better model (i.e., student model). Due to its simplicity, many attempts have been made for the field of semi-supervised learning [18, 2, 4, 28] and unsupervised domain adaptation [49, 50]. Moreover, recent methods [39, 46] show that self-training can surpass state-of-the-art fully-supervised models on large-scale datasets such as ImageNet.

However, most of these works focus on self-training for classification or object detection tasks, while self-training for tasks considering geometric predictions (e.g., room-layout estimation studied in this paper) is rarely explored. SSLayout360 [34] trains a layout estimation model with pseudo-labels produced by the Mean Teacher [33]. Yet, the simple extension from classification to layout estimation considers each image independently, ignoring the geometry information coming from other camera views. In addition, another challenge is that pseudo-labels are usually noisy. Previous methods attempt to reduce the noise by ensembling multiple predictions for an image under different augmentations [4, 22] or by selecting only the pseudo-labels with high confidence [28]. In this paper, we leverage multi-view consistency from the layout estimations and measure their uncertainty to construct reliable pseudo-labels.

**Unsupervised model validation.** Without ground truth annotations, how to evaluate a machine learning model remains an open issue. Traditional unsupervised learning algorithms (e.g., clustering) can be evaluated without external labels through computing cohesion and separation [32]. For unsupervised domain adaptations, where the problem assumes no labeled data is available in the target domain, unsupervised validation is more practical for hyper-parameter tuning [20, 25]. [20] evaluates the confidence of the predictions of the classifier using entropy. [25] proposes the soft neighborhood density, measuring the local similarity between data samples (the higher, the better). However, these metrics are designed for classification or segmentation tasks and cannot easily be adopted by our tasks. Therefore, in this paper, we study unsupervised validation from another perspective: evaluating the geometry consistency between multiple views, requiring no ground truth annotations.

# 3   Our Approach: 360-MLC

Our primary goal is to deploy a model pre-trained on source domain into a new dataset (target domain), where the data distribution may differs from the one used in the pre-trained model. We assume that several images in the new scene are captured and registered by their camera poses, but their layout ground truth annotations are not available. Under this practical scenario, we present 360-MLC, a self-training method that is based on multi-view layout consistency.

For illustration purposes, Fig. 1 presents the overview of our method. First, we begin with a set of 360-images and a pre-trained room-layout model that can generate a set of estimated layout boundaries (see green lines in Fig. 1-(a)). Then, by leveraging the related camera poses of every image, all layout boundaries are projected into the world coordinate and then re-projected them back into a target view (see yellow lines in Fig. 1-(b)). Details of these projections are described in §3.1.

Assuming that the projected boundaries from all the views describe the same scene geometry, we can compute the most likely layout boundary positions as pseudo-labels for self-training along with their uncertainty based on the variance (see Fig. 1-(c)).   Upon the estimated pseudo-labels, we define our weighted boundary consistency loss $\mathcal{L}_{\mathrm{WBC}}$, allowing us to define a reliable regularization used for self-training. More details are presented in §3.2.

A challenging step towards our goal is the lack of a metric for evaluating a layout estimator when no ground truth annotations are available. To tackle this issue, §3.3 describes our multi-view layout consistency metric $\mathcal{H}_{\mathrm{MLC}}$, which allows us to evaluate multiple layout estimates without requiring any ground truth. We leverage the proposed metric throughout all the experiments described in §4 as the quantitative metric for model selection and hyper-parameter tuning.

## 3.1   Multi-view Layout Re-projection

In this section, we describe the multi-view layout re-projection process of all estimated layout boundaries from different cameras positions in the scene. To this end, we define the set of 360-images and boundaries in the scene as follows:

$$\{(I_i, Y_i)\}_{i=1:N}, \quad I_i \in \mathbb{R}^{H \times W}, \quad Y_i \in \mathbb{R}^W , \tag{1}$$

where $I_i$ is the i-$th$ 360-image with the size of $W$ columns times $H$ rows pixels, $Y_i$ is the layout boundary of image $I_i$, and $N$ is the number of images. $Y_i \in \mathbb{R}^W$ is a vector of boundaries at all columns, where $Y_i(\theta) = \phi$ specifies that the layout boundary is at row $\phi$ for column $\theta$ in the pixel coordinate.

For a supervised training, $Y_i$ is given as a ground truth label. However, in our proposed self-training framework, we aim to ensemble pseudo-labels through geometry re-projection of multiple views. To begin with, we describe the process to project the boundary $Y_i$ into the j-$th$ target view as the re-projected boundary $Y_{i \to j}$. The projection of $Y_i$ into the world coordinate using the Euclidean space is described as follows:

$$X_i = \mathrm{Proj}(Y_i, T_i, h_i) , \tag{2}$$

where $X_i$ is the projected layout boundary in the world coordinate; $h_i$ is the camera height; $T_i \in \mathrm{SE}(3)$ is the camera pose with respect to the world coordinate; $\mathrm{Proj}(\cdot)$ is the layout projection for spherical cameras that maps spherical coordinates $[\theta, \phi]^\top \in \mathbb{R}^2$ into the 3D world coordinate $[x, y, z]^\top \in \mathbb{R}^3$. Details of this projection function $\mathrm{Proj}(\cdot)$ and how we handle unknown camera heights with estimated poses are presented in the supplementary material.

Then, $X_i$ in the world coordinate can be re-projected into a j-$th$ target view ($Y_{i \to j}$) as follows:

$$Y_{i \to j} = \mathrm{Proj}^{-1}(T_j, X_i) , \tag{3}$$

where $\mathrm{Proj}^{-1}(\cdot)$ is the inverse of the projection function presented in Eq. (2). We collect all re-projected boundaries into $\mathbf{Y}_j = [Y_{1 \to j}; \ldots; Y_{N \to j}] \in \mathbb{R}^{W \times N}$, where $\mathbf{Y}_j(\theta) \in \mathbb{R}^N$ is a vector of row positions on boundaries from $N$ views at column $\theta$. As shown in the yellow lines in Fig. 1-(b), the visualization of $\mathbf{Y}_j$ reveals the underlying geometry of the scene. More visualization results are presented in §4.2.

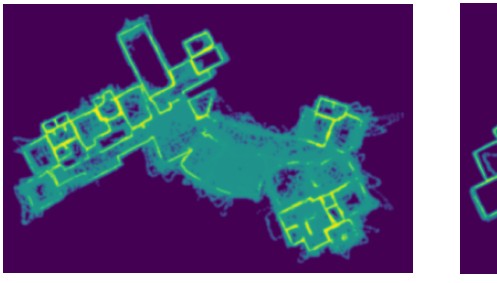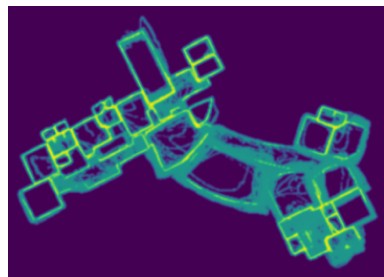

|(a) Pre-trained model|(b) After our self-training|

Figure 2: **2D density function of multi-view layouts.** We project multiple layout boundaries into a top-view 2D density map that reveals the geometry consistency in the scene. (a) We present the projection of multi-view layout using a pre-trained model before self-training. (b) We observe a better layout consistency of the scene after our self-training process. In §3.3, we use such a top-view map for calculating our layout consistency metric $\mathcal{H}_{\text{MLC}}$.

## 3.2 Weighted Boundary Consistency Loss

In this section, we describe how multiple re-projected boundaries can be used to estimate pseudo-label boundaries and its uncertainty as a reliable supervision for our self-training formulation. First, we obtain a set of predicted boundaries $\{Y_i = \mathcal{M}(I_i)\}_{i \in 1:N}$ for all views using the pre-trained model $\mathcal{M}$. Following the process in §3.1, we re-project the estimated boundaries into $\mathbf{Y}_j$ for each view. We define the pseudo-label ($\bar{Y}_j$) and its uncertainty ($\sigma_j$) as follows:

$$\bar{Y}_j = [\text{Median}(\mathbf{Y}_j(\theta))]_{\theta=1:W} \in \mathbb{R}^W , \quad \sigma_j = [\text{STD}(\mathbf{Y}_j(\theta))]_{\theta=1:W} \in \mathbb{R}^W , \quad (4)$$

where Median($\cdot$) and STD($\cdot$) are the median and standard deviation both applied to $\mathbf{Y}_j(\theta)$ (i.e., $N$ re-projections of row positions on boundary at column $\theta$). The pseudo-labels are computed for each target view. By leveraging Eq. (4), we define our geometry loss as follows:

$$\mathcal{L}_{\text{WBC}} = \sum_{j=1}^{N} \sum_{\theta=1}^{W} \frac{||Y_j(\theta) - \bar{Y}_j(\theta)||_1}{\sigma_j^2(\theta)} . \quad (5)$$

$\mathcal{L}_{\text{WBC}}$ is our proposed weighted boundary consistency loss for finetuning the room-layout model $\mathcal{M}$. Note that the denominator weights the pseudo-label accordingly as the uncertainty at each column $\theta$. This design aims to reduce the effect of unstable predictions in the scene (e.g., drastic occlusions) that generally incur a larger variance in the re-projection due to the inconsistency between multiple views.

## 3.3 Multi-view Layout Consistency Metric

In practice, one challenge for adapting a pre-trained model into a completely unlabeled new data domain is that there is no labelled hold-out dataset for tuning hyper-parameters such as learning rate, etc. Hence, we need a metric to measure the performance based on unlabeled data. To this end, we propose a the $\mathcal{H}_{\text{MLC}}$ metric to measure the multi-view layout consistency that does not rely on ground truth labels but on model outputs only.

First, we collect all estimated layout boundaries in the scene projected by Eq. (2) to build a top-view 2D density map. This can be described as follows:

$$\mathbf{X} = \{X_i\}_{i=1:N}, \quad \Phi(\mathbf{X}) \in \mathbb{R}^{U \times V}, \quad (6)$$

where $\mathbf{X}$ is the set of all estimated layouts in the scene, and $\Phi(\cdot)$ is a top-down projection function that maps $\mathbf{X}$ into a discrete 2D-grid with the size $U \times V$ as a normalized histogram. Our key idea is to evaluate the entropy in this discrete grid as follows:

$$\mathcal{H}_{\text{MLC}} = \sum_{u,v} -\Phi_{u,v}(\mathbf{X}) \cdot \log \Phi_{u,v}(\mathbf{X}). \quad (7)$$

The intuition behind this evaluation comes from the fact that better alignment of layout boundaries yield in a lower entropy evaluation, while a higher entropy value would reflect the poor alignment of

Table 1: Dataset statistics.

| Source Dataset | Number of Frames | Target Dataset | Number of Frames |
|---|---|---|---|
| MatterportLayout [48] | 2094 | MP3D-FPE [29] | 2094 |
| ZInD [9] | 2094 | MP3D-FPE | 2094 |
| LayoutNet [47] | 817 | MP3D-FPE | 817 |

layout geometry between multiple views. As a result, we can use $\mathcal{H}_{\mathrm{MLC}}$ to select hyper-parameters and early stop the model training, without using any ground truth annotations. For illustration purposes, Fig. 2 presents two projected scenes that show how the geometry consistency correlates with a less disordered 2D projection. More details are presented in the supplementary material.

## 4 Experiments

### 4.1 Experimental Setup

**Datasets.** We conduct extensive experiments using publicly available 360-image layout datasets: Matterport3D Floor Plan Estimation (MP3D-FPE) [29] as the target dataset, and three real-world datasets as the pre-training datasets, including MatterportLayout [36, 48], Zillow Indoor Dataset (ZInD) [9], and the dataset used in LayoutNet [47] that combines PanoContext [45] and Stanford2D3D [1] (referred to as the LayoutNet dataset for simplicity). MP3D-FPE is collected using the MINOS simulator [26] to render sequences of 360-images within each scene/room from the Matterport3D dataset [6]. Note that it is the only dataset containing multi-view 360-images and thus we consider it as the target dataset.

In experiments, we aim to pre-train the layout model using each of the pre-trained dataset, and then we utilize the target MP3D-FPE dataset for self-training and performing evaluation. We follow the standard training split released in each pre-trained dataset. For the target dataset MP3D-FPE, we use 2094 frames as the training set and 700 frames as the testing set, from 32 and 7 scenes, respectively, where those scenes are not included in the training set of the Matterport3D dataset. On average, there are 10.46 views within each room. Note that the target MP3D-FPE dataset is a challenging dataset since it includes many complex scenes that do not follow the Manhattan assumption [8], and we carefully label the ground truth 2D and 3D layout for the testing set. As a result, compared to other datasets, the performance scores on MP3D-FPE are lower when the model is evaluated using state-of-the-art methods.

**Evaluation metrics.** We follow Zou *et al.* [48] to construct the four standard protocols for evaluation. To evaluate layout boundary, we use 2D and 3D intersection-of-union (IoU). To evaluate layout depth, we use root-mean-square error (RMSE) by setting the camera height as 1.6 meters, and $\delta_1$, which describes the percentage of pixels where the ratio between the estimation and the ground truth depth is within the threshold of 1.25. In addition, we introduce a new metric $\mathcal{H}_{\mathrm{MLC}}$ outlined in §3.3 to analyze the consistency between multiple layout estimations. We show in §4.3 that $\mathcal{H}_{\mathrm{MLC}}$ is highly co-related to 2D/3D IoUs on the hold-out dataset. Hence, $\mathcal{H}_{\mathrm{MLC}}$ is suitable for hyper-parameter tuning when ground truths are not available.

**Implementation details.** We adopt HorizonNet [30] as our layout estimation backbone due to its state-of-the-art performance. Note that, different from the original HorizonNet, our model is trained without the supervision on the corner channel since our pseudo-label contains only the boundary information. Common data augmentation techniques for 360-images are also applied during training, including left-right flipping, panoramic horizontal rotation, and luminance augmentation. We use the Adam optimizer to train the model for 300 epochs by setting the learning rate as 0.0001 and the batch size as 4. We save the model every 5 epochs and the early-stopped model is selected based on the lowest $\mathcal{H}_{\mathrm{MLC}}$ score on the hold-out test dataset for all methods. All models are trained on a single NVIDIA TITAN X GPU with 12 GB of memory. We will make our models, codes, and dataset available to the public.

Table 2: Evaluation results of Setting 1 on MP3D-FPE [29].

| Pre-trained Dataset | Method | 2D IoU (%) ↑ | 3D IoU (%) ↑ | RMSE ↓ | $\delta_1$ ↑ | $\mathcal{H}_{\text{MLC}}$ ↓ |
|---|---|---|---|---|---|---|
| MatterportLayout [48] | Pre-trained | 65.38 | 62.28 | 0.58 | 0.78 | 8.18 |
| | SSLayout360* [34] | 70.53 | 66.74 | 0.48 | 0.82 | 8.15 |
| | Ours | **71.50** | **67.70** | **0.46** | **0.82** | **8.10** |
| ZInD [9] | Pre-trained | 45.43 | 42.17 | 1.02 | 0.61 | 8.32 |
| | SSLayout360* | 62.62 | 58.27 | 0.66 | 0.74 | 8.34 |
| | Ours | **65.60** | **60.70** | **0.56** | **0.75** | **8.19** |
| LayoutNet [47] | Pre-trained | 64.34 | 58.92 | 0.61 | 0.70 | 8.50 |
| | SSLayout360* | 67.48 | 62.89 | 0.56 | **0.77** | **8.29** |
| | Ours | **69.40** | **65.22** | **0.55** | 0.72 | 8.32 |

Table 3: Evaluation results of Setting 2 on MP3D-FPE [29].

| Pre-trained Dataset | Method | 2D IoU (%) ↑ | 3D IoU (%) ↑ | RMSE ↓ | $\delta_1$ ↑ | $\mathcal{H}_{\text{MLC}}$ ↓ |
|---|---|---|---|---|---|---|
| MatterportLayout [48] | SSLayout360-ST* | 70.58 | 66.64 | 0.52 | **0.81** | **8.10** |
| | Ours | **71.50** | **67.20** | **0.49** | 0.78 | 8.14 |
| ZInD [9] | SSLayout360-ST* | 56.52 | 52.12 | 0.72 | 0.74 | 8.20 |
| | Ours | **64.45** | **59.53** | **0.59** | **0.75** | **8.17** |
| LayoutNet [47] | SSLayout360-ST* | 66.14 | 61.51 | 0.57 | **0.76** | **8.30** |
| | Ours | **69.12** | **64.62** | 0.57 | 0.69 | 8.31 |

## 4.2 Experimental Results

**Baselines.** The proposed framework is compared against our re-implementation of SSLayout360 [34]. We ensure that our re-implemented SSLayout360* has similar performance compared to the reported results in [34]. Note that we use the same backbone, i.e., HorizonNet, for both our 360-MLC and SSLayout360*, and we train models using only the boundary supervision as stated in §4.1. We implement an extended self-training version based on SSLayout360*, namely SSLayout360-ST*, in which we disable the supervised loss and train without any ground truth annotations. We initialize all models with the same pre-trained weights from the official HorizonNet[1] release. In addition, different from our approach, both SSLayout360* and SSLayout360-ST* are trained on two NVIDIA TITAN X GPUs due to the requirement of larger memory for their teacher-student architecture.

**Setting 1: labeled pre-training data + unlabeled target data.** In this setting, we sample the same amount of training data from both the pre-training dataset and the target dataset, as shown in Table 1. We start from the pre-trained model, and then generate pseudo-labels for unlabeled data. Then, during model finetuning, we include both labeled pre-trained data and pseudo-labeled data from MP3D-FPE. In Table 2, we show that our method consistently performs favorably against SSLayout360* across most metrics, in which SSLayout360* does not consider the usage of multi-view layout consistency as our approach does.

**Setting 2: pre-trained model + unlabeled target data.** A more practical and challenging setting is that only the pre-trained model and unlabeled data in the target domain are available during model finetuning. In Table 3, similar to Setting 1, our method has consistent performance improvement against SSLayout360-ST*. Note that we observe that our performance gains are larger (especially on ZInD and LayoutNet) compared to Setting 1. For MatterportLayout, since its data distribution is closer to the MP3D-FPE (both are from Matterport3D), the performance difference is less.

Moreover, when we remove the labeled pre-trained data from Setting 1, we do not observe a significant performance drop in Setting 2 on all dataset settings, while SSLayout360-ST* is more sensitive to the labeled pre-trained data, e.g., on ZInD, 2D/3D IoU is decreased by 6.1%/6.15%. This demonstrates the

---

[1]https://github.com/sunset1995/HorizonNet

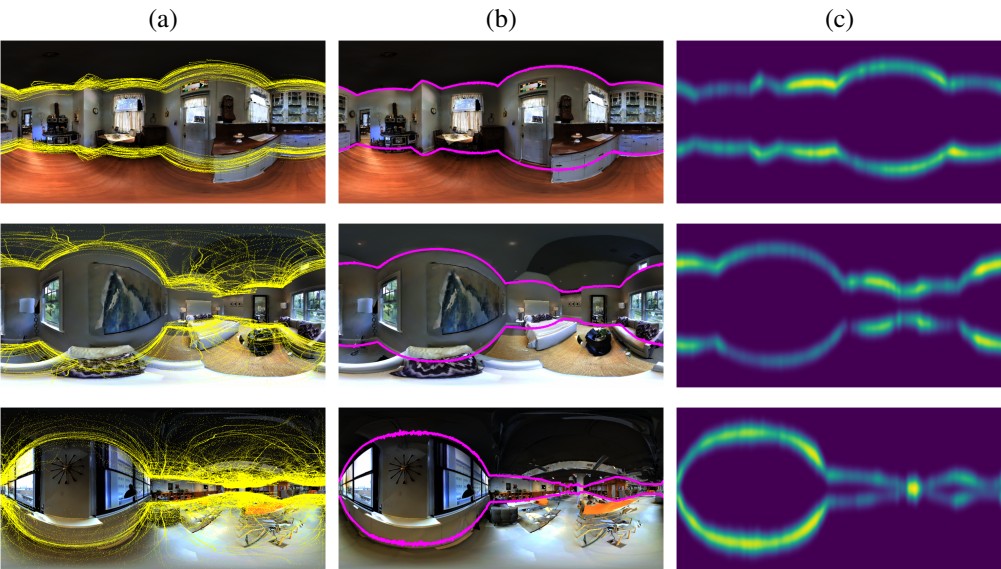

| (a) | (b) | (c) |

Figure 3: **Multi-view pseudo-labeling.** We show the qualitative visualization of our proposed 360-MLC. In (a), all re-projected layout boundaries from different camera views are presented as yellow lines. In (b), the corresponded pseudo-labels are depicted in magenta. In (c), the uncertainty in the pseudo labels is shown as 2D maps. We can appreciate that our proposed 360-MCL can estimate plausible pseudo-labels from estimations along the scene.

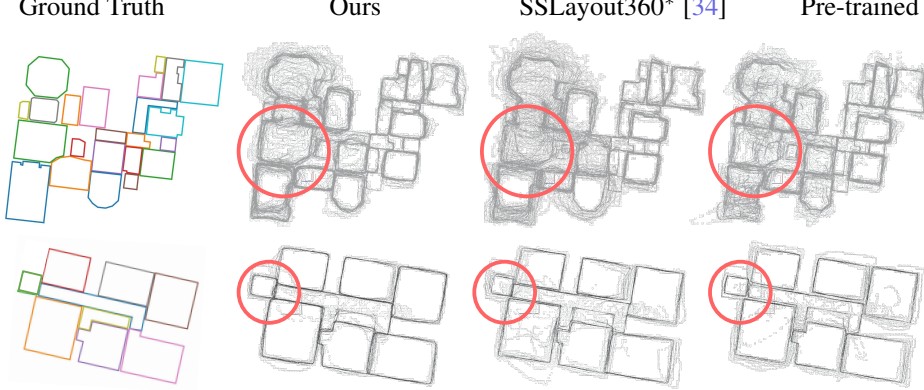

| Ground Truth | Ours | SSLayout360* [34] | Pre-trained |

Figure 4: **Qualitative results in the 2D top-view.** We project the estimated layouts registered by the corresponding camera poses into 2D. Comparing to our baseline SSLayout360* [34] and the pre-trained model, our model produces more consistent layouts (highlighted with red circles).

effectiveness of considering multi-view layout consistency that generates more reliable pseudo-labels, even when no labeled data is provided during training.

**Qualitative results.** For illustration purpose, we visualize the proposed multi-view pseudo-labeling in Fig 3. In Fig. 4 we show qualitative results of multiple layouts projected into the 3D world coordinate for two test scenes evaluated on MP3D-FPE [29], following the Setting 2. It can be observed that our proposed 360-MLC presents sharper and clearer layout boundaries for those scenes, demonstrating a better performance compared with the baselines. Lastly, we analysis the qualitative layout predictions on 360-images in Fig 5.

## 4.3 Ablation Study

In this section, we present our ablation study to validate the effectiveness of the proposed components. We conduct experiments using the ZInD pre-trained weights under the Setting 2 mentioned in §4.2.

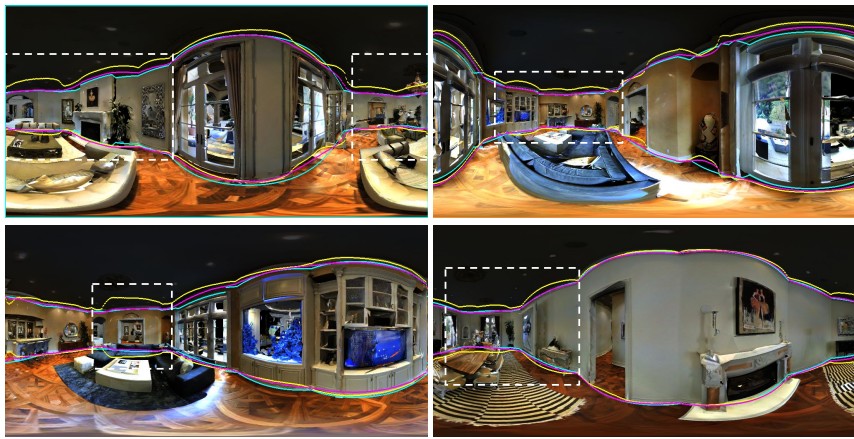

Figure 5: **Qualitative comparisons on 360-images.** We compare our model with SSLayout360* in 360-images. The cyan, yellow, and magenta lines are ground truth, SSLayout360*, and 360-MLC, respectively. We observe that our model predicts layout boundary more closely to the ground truth than SSLayout360*, and is more robust towards furniture (e.g., tables, couch) and large rooms. The dashed white bounding boxes highlight the error predictions from the baseline.

Table 4: Ablation study with Setting 2 on MP3D-FPE [29], pre-trained on ZInD [9].

|     | Frames (%) | Median | Mean | $\mathcal{L}_{\text{WBC}}$ | 2D IoU (%) ↑ | 3D IoU (%) ↑ | RMSE ↓ | $\delta_1$ ↑ | $\mathcal{H}_{\text{MLC}}$ ↓ |
|-----|-----------|--------|------|------|------------|------------|--------|--------|--------|
| (a) | 10  | ✓ | -  | ✓ | 57.29 | 53.44 | 0.71 | 0.68 | 8.23 |
| (b) | 50  | ✓ | -  | ✓ | 59.55 | 55.46 | 0.65 | 0.69 | 8.22 |
| (c) | 100 | -  | ✓  | ✓ | 46.17 | 42.63 | 0.82 | 0.63 | 8.22 |
| (d) | 100 | ✓ | -  | - | 62.81 | 58.05 | **0.58** | **0.76** | 8.18 |
| (e) | 100 | ✓ | -  | ✓ | **64.45** | **59.53** | 0.59 | 0.75 | **8.17** |

Results are presented in Table 4, describing four main experiments. First, we investigate how the number of views used to generate our pseudo-labels may affect the performance of our proposed solution. Second, we replace the Median function in Eq. (4) with Mean, aiming to demonstrate the effects of outliers in the quality of our pseudo-labels. Then, we analyze the proposed $\mathcal{L}_{\text{WBC}}$ with the vanilla $\mathcal{L}_1$ loss function. Lastly, we show how $\mathcal{H}_{\text{MLC}}$ could be used in hyper-parameter tuning, such as selecting learning rates and models.

**Multi-view pseudo-labeling.** In this experiment, we aim to verify whether the more views of estimations we use, the better quality of the pseudo-labels we can acquire. Therefore, we experiment three models trained with the same amount of data but using pseudo-labels created by 10%, 50% and 100% of frames, as shown in row (a), (b), and (e), respectively in Table 4. The result demonstrates the contribution of our proposed method: using the layout estimations from more views for self-training can help ensemble more robust training signals.

**Median and mean in Eq. (4).** In this experiment, we investigate the impact of the median operator for pseudo-label generation. We compare the mean and median functions in row (c) and (e) in Table 4. It can be observed that the median function has a better performance since it is capable of ignoring outliers in the re-projected boundaries from multiple views, increasing the robustness of pseudo-labels.

**$\mathcal{L}_{\text{WBC}}$ and $\mathcal{L}_1$.** The results in row (d) and (e) of Table 4 show that adding the uncertainty $\sigma$ in $\mathcal{L}_{\text{WBC}}$ performs better than the simple $\mathcal{L}_1$ loss. This is because our proposed loss function down-weights the part of the layout boundaries that are noisy (i.e., high variance), avoiding the effect of unreliable pseudo-labels during self-training.

**$\mathcal{H}_{\text{MLC}}$ for hyper-parameter tuning.** We present an example of using our proposed metric $\mathcal{H}_{\text{MLC}}$ for hyper-parameter tuning. In Fig. 6, we test three different settings using the same model training

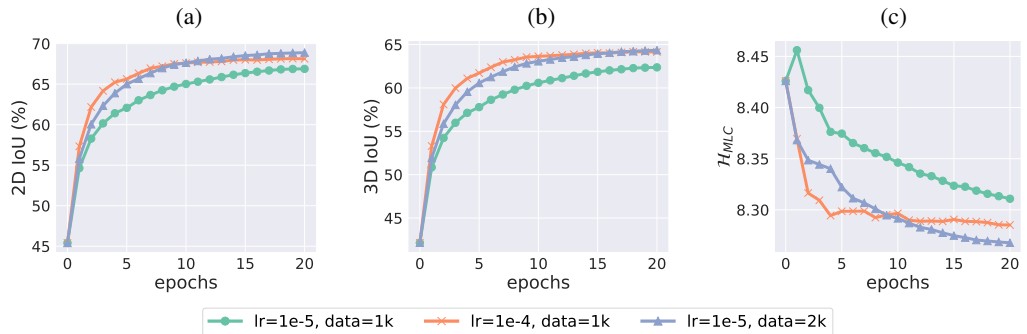

Figure 6: **An example of hyper-parameter tuning.** We show that the layout evaluation metrics, (a) 2D IoU and (b) 3D IoU, evaluated using ground truth labels, are inversely correlated with the proposed metric (c) $\mathcal{H}_{\text{MLC}}$ under three different settings. In this way, we are able to select better models with the lower value of $\mathcal{H}_{\text{MLC}}$. We conduct the experiment under two conditions: different learning rates and different amount of training data.

Table 5: Evaluation results of Setting 2 on MP3D-FPE [29] using estimated poses.

| Pre-trained Dataset | Method | 2D IoU (%) ↑ | 3D IoU (%) ↑ | RMSE ↓ | $\delta_1$ ↑ |
|---|---|---|---|---|---|
| MatterportLayout [48] | Ours + ground truth poses | **71.50** | **67.20** | 0.49 | **0.78** |
| | Ours + estimated poses | 70.85 | 66.91 | **0.48** | **0.78** |
| | Ours + noisy poses | 66.05 | 61.41 | 0.71 | 0.65 |

process. Among them, using learning rate $1 \times 10^{-5}$ and $1K$ training samples (green lines) performs the worst, while adopting learning rate $1 \times 10^{-5}$ and $2K$ training samples (blue lines) performs the best. We show that the trend of our proposed unsupervised metric (Fig. 6-(c)) is consistent with the two supervised metrics, 2D/3D IoU (Fig. 6-(a) and (b)). Therefore, even without ground truth labels, our metric can be served as a robust indication for validation.

## 5 Limitations

Several views from the same scene with their registered camera poses are required to formulate the proposed 360-MLC. Although the registration of multiple camera poses can be accomplished accurately by external sensors, structure from motion (SfM), or Simultaneous Localization and Mapping (SLAM) solutions, any error in this registration may lead to poor performance. To complement the experiments depicted in §4, Table 5 shows that our proposed method can keep similar performance using estimated poses under mild noise conditions. However, under severe noise conditions, we can appreciate a lower performance for our proposed solution.

## 6 Conclusions

We present 360-MLC, a self-training method based on multi-view layout consistency for finetuning monocular 360-layout models using unlabeled data only. Our method tackles a practical scenario where a pre-trained model needs to be adapted to a new data domain without using any ground truth annotations. In addition, we leverage the entropy information in multiple layout estimations as a quantitative metric to measure the geometry consistency of the scene, allowing us to evaluate any layout estimator for hyper-parameter tuning and model selection in an unsupervised fashion. Experimental results show that our self-training solution achieves favorable performance against state-of-the-art methods from three publicly available source datasets to the newly labeled multi-view MP3D-FPE dataset.

## 7 Acknowledgements

This work is supported in part by Ministry of Science and Technology of Taiwan (MOST 110-2634-F-002-051). We thank National Center for High-performance Computing (NCHC) for computational and storage resource.

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
