# Supplementary

## 1  Outline

The following is the outline of this supplementary material:

- In appendix A, we present our upper-bound experiments which complements the experiments presented in Table 2 and Table 3 of our main manuscript.
- In appendix B, we present extra qualitative results to show the advantage of our proposed method.
- In appendix C, we present additional details of the camera projection model described by Eq. (2) and Eq. (3) in our main manuscript.
- In appendix D, we present details about the re-implementation of SSLayout360 and its results.
- In Figure 1, we show a t-SNE plot of the distributions in four datasets used in this work.

## A  Upper-bound Results

Table 1: Evaluation results of Setting 1 on MP3D-FPE [2].

| Pre-trained Dataset | Method | 2D IoU (%) ↑ | 3D IoU (%) ↑ | RMSE ↓ | $\delta_1$ ↑ |
|---|---|---|---|---|---|
| MatterportLayout [7] | Pre-trained | 65.38 | 62.28 | 0.58 | 0.78 |
| | SSLayout360* [4] | 70.53 | 66.74 | 0.48 | **0.82** |
| | Ours | **72.38** | **68.16** | **0.51** | 0.79 |
| ZInD [1] | Pre-trained | 45.43 | 42.17 | 1.02 | 0.61 |
| | SSLayout360* | 62.59 | 58.17 | 0.65 | 0.74 |
| | Ours | **66.63** | **61.88** | **0.57** | **0.75** |
| LayoutNet [6] | Pre-trained | 64.34 | 58.92 | 0.61 | 0.70 |
| | SSLayout360* | 66.99 | 62.50 | 0.55 | **0.78** |
| | Ours | **70.02** | **63.59** | **0.64** | 0.67 |

In Table 1 and Table 2 of this supplementary material, we present our upper-bound results that complement the experiments shown in Table 2 and Table 3 of the main manuscript. These upper-bound experiments use the same settings described in Sec 4 of our main manuscript, but instead of using the lowest entropy $\mathcal{H}_{\text{MLC}}$ for model selection, here we use the best 2D IoU evaluation. Therefore, these experiments represent the scenario when ground truth annotations are available during testing.

Comparing the results presented in our main manuscript with the ones depicted here, we can verify that indeed our proposed metric $\mathcal{H}_{\text{MLC}}$ can lead to similar best models but without using ground truth annotations. In addition, our proposed method still outperforms other baselines, for both settings (i.e.,

Table 2: Evaluation results of Setting 2 on MP3D-FPE [2].

| Pre-trained Dataset | Method | 2D IoU (%) ↑ | 3D IoU (%) ↑ | RMSE ↓ | $\delta_1$ ↑ |
|---|---|---|---|---|---|
| MatterportLayout [7] | SSLayout360-ST* | 70.07 | 66.19 | **0.47** | **0.82** |
| | Ours | **72.67** | **67.72** | 0.53 | 0.75 |
| ZInD [1] | SSLayout360-ST* | 55.18 | 51.18 | 0.71 | **0.74** |
| | Ours | **67.62** | **62.42** | **0.56** | **0.74** |
| LayoutNet [6] | SSLayout360-ST* | 66.14 | 61.51 | **0.57** | **0.76** |
| | Ours | **68.78** | **61.74** | 0.70 | 0.64 |

Setting 1 and 2 described in Sec 4.2 in our main manuscript), which demonstrates the efficacy of our self-training formulation.

# B  Qualitative Results

In this section, we present additional qualitative visualizations of multi-view layout consistency projected to the top-view via the 2D density function in Figure 2.

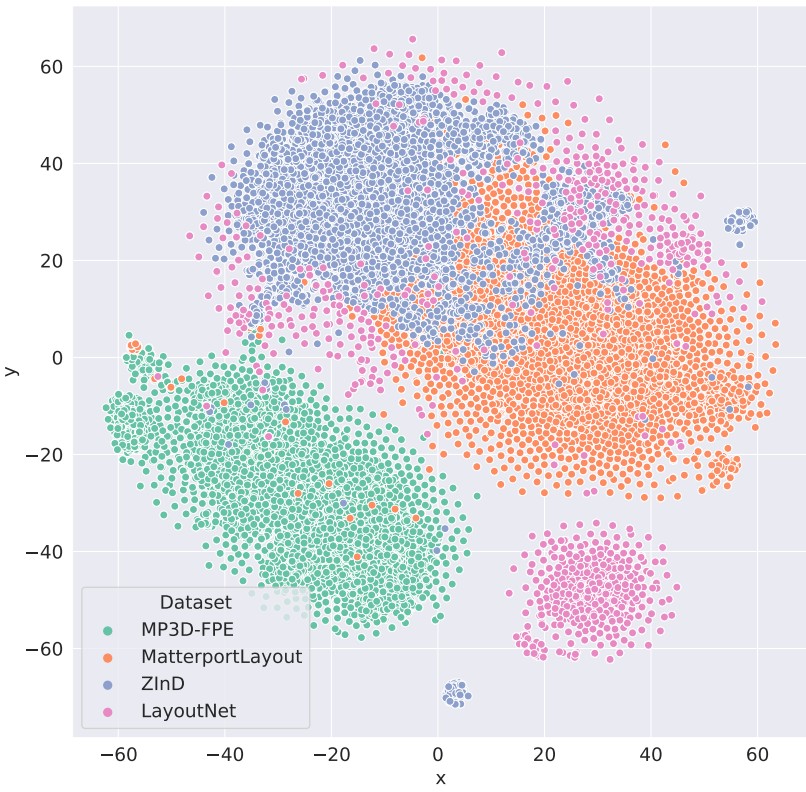

Figure 1: **t-SNE plot of the four datasets in this work.** It is observed that the three source datasets, i.e., MatterportLayout, ZInD, and LayoutNet, have different data distributions with respect to the target dataset, MP3D-FPE [2]. This is because these source datasets consist of real-world images, whereas our target dataset is rendered from a simulator.

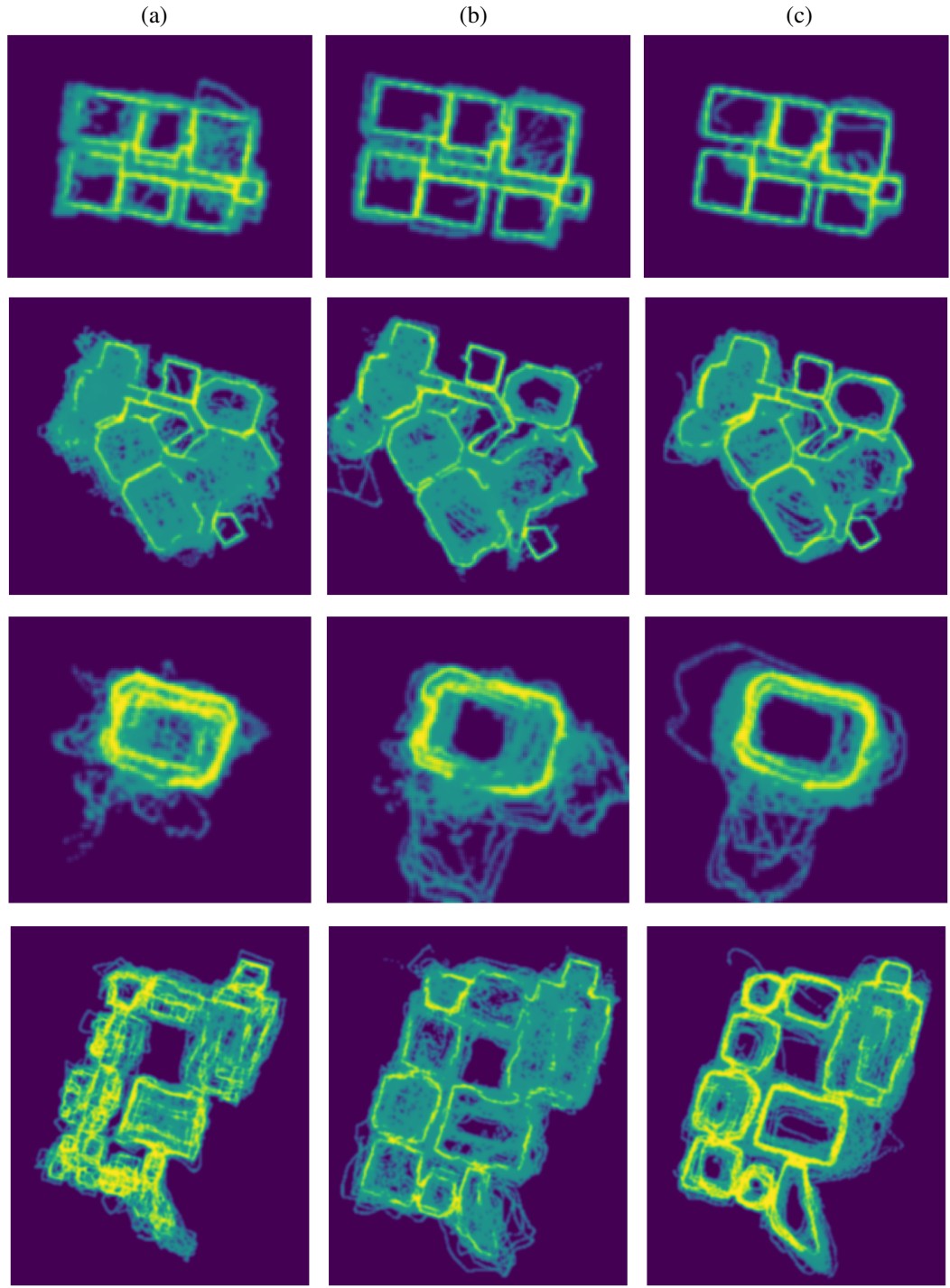

Figure 2: **2D density function of multi-view layouts.** We show the qualitative results on MP3D-FPE dataset [2] for a pre-trained model on ZinD dataset [1] (a), the baseline SSLayout360* [4] (b), and our proposed 360-MLC (c). In this figure, a sharper yellow silhouette represents an accumulation of multiple layout estimations, hence showing the geometry consistency along the scene.

## C  Camera Projection Model

Let

$$\text{Proj}: \mathbb{R}^2 \rightarrow \mathbb{R}^3, \quad (\theta, \phi) \rightarrow (x, y, z) \tag{1}$$

denote the $\text{Proj}(\cdot)$ function introduced in Eq. (2) of the main paper, which maps the spherical coordinate $(\theta, \phi) \in \mathbb{R}^2$ to 3D world coordinate $(x, y, z) \in \mathbb{R}^3$. We can expand this function as follows:

$$\mathbf{X}_i = \text{Proj}\big(Y_i, T_i, h_i\big) = \bar{\mathbf{X}}_i \cdot \mathbf{R}_i + \mathbf{t}_i, \quad \bar{\mathbf{X}}_i = \begin{bmatrix} x \\ y \\ z \end{bmatrix} = \begin{bmatrix} \frac{|h_i|}{\sin \phi} \cos \phi \sin \theta \\ h_i \\ \frac{|h_i|}{\sin \phi} \cos \phi \cos \theta \end{bmatrix}, \tag{2}$$

where $\mathbf{X}_i \in \mathbb{R}^3$ is the layout projected in world coordinates; $Y_i$ is the layout boundary (cf. Sec. 3.1); $T_i = [\mathbf{R}_i | \mathbf{t}_i]$ is the camera pose for i-$th$ camera view parameterized by the rotation matrix $\mathbf{R}_i \in SO(3)$ and translation vector $\mathbf{t}_i \in \mathbb{R}^3$; $h_i$ is the camera height; $\bar{\mathbf{X}}_i$ is the projected layout in the i-$th$ camera coordinates. Note that every pair of spherical coordinates $(\theta, \phi)$ is first projected into 3D Euclidean space as $\bar{\mathbf{X}}_i \in \mathbb{R}^3$ in camera reference and then registered into world coordinates by $T_i$.

In the case of projecting the floor boundary, the camera height is defined as $h_i = h_i^{(f)}$, i.e., the distance from the camera center to the floor. However, for projecting the ceiling boundary, the camera height $h_i = h_i^{(c)}$ can be computed by $h_i^{(f)}$ given the assumption that walls are perpendicular to the floor and ceiling [3], which is defined as follows:

$$h_i^{(c)} = \frac{1}{W} \sum_{\theta=1}^{W} -h_i^{(f)} \cot Y_i^{(f)}(\theta) \cdot \tan Y_i^{(c)}(\theta), \tag{3}$$

where $Y_i^{(f)}$ and $Y_i^{(c)}$ are the layout boundaries for floor and ceiling respectively, $W$ is the number of column defined in $Y_i$ (cf. Sec. 3.1), and $h_i^{(c)}$ is the distance from the camera center to the ceiling.

We can detail the back-projection function $\text{Proj}^{-1}(\cdot)$ as follows:

$$\text{Proj}^{-1}: \mathbb{R}^3 \rightarrow \mathbb{R}^2, \quad (x, y, z) \rightarrow (\theta, \phi), \tag{4}$$

$$Y_{i \rightarrow j} = \text{Proj}^{-1}\big(T_j, \mathbf{X}_i\big), \tag{5}$$

$$\bar{\mathbf{X}}_j = \mathbf{X}_i \cdot \mathbf{R}_j^\top - \mathbf{R}_j^\top \cdot \mathbf{t}_j, \quad \begin{bmatrix} x \\ y \\ z \end{bmatrix} = \frac{\bar{\mathbf{X}}_j}{\|\bar{\mathbf{X}}_j\|_2}, \tag{6}$$

$$\begin{bmatrix} \theta \\ \phi \end{bmatrix} = \begin{bmatrix} \tan^{-1}\big(\frac{x}{z}\big) \\ \sin^{-1}(-y) \end{bmatrix}, \quad \forall\, Y_{i \rightarrow j}(\theta) = \phi, \tag{7}$$

where $\mathbf{X}_i$ is the layout geometry projected by Eq. (2), $T_j$ is the camera pose of the j-$th$ image view parameterized by the rotation matrix $\mathbf{R}_j \in SO(3)$ and translation vector $\mathbf{t}_j \in \mathbb{R}^3$, $\bar{\mathbf{X}}_j$ is the geometry layout in the j-$th$ camera reference, and $Y_{i \rightarrow j}$ is the layout boundary in spherical coordinates at the j-$th$ references. Here, every set of coordinates $(x, y, z) \in \mathbb{R}^3$ defined by $\mathbf{X}_i$ is first transformed in j-$th$ camera reference by $T_j$ and then mapped into spherical coordinates by Eq. (7).

## D  SSLayout360 and SSLayout360*

In Table 3, we present the results of our re-implemented SSLayout360* along with the original SSLayout360 [4] on the MatterportLayout [7, 5] dataset. We follow the standard training, validation, and testing splits from Zou *et al.* [7] and the list of labeled subset provided by SSLayout360[1]. We show that our implementation has comparable performance with the official results presented by SSLayout360. For more implementation details, please refer to [4].

---

[1] https://github.com/FlyreelAI/sslayout360

Table 3: Quantitative results of SSLayout360 and SSLayout360* on MatterportLayout [7].

| Labels / Images | Method | 2D IoU (%) ↑ | 3D IoU (%) ↑ | RMSE ↓ | $\delta_1$ ↑ |
|---|---|---|---|---|---|
| 50 / 1837 | SSLayout360 | 71.03 | 67.42 | 0.35 | 0.81 |
| | SSLayout360* | 69.95 | 66.10 | 0.34 | 0.70 |
| 100 / 1837 | SSLayout360 | 75.46 | 72.37 | 0.29 | 0.89 |
| | SSLayout360* | 75.26 | 71.95 | 0.26 | 0.80 |
| 200 / 1837 | SSLayout360 | 78.05 | 75.31 | 0.27 | 0.91 |
| | SSLayout360* | 77.87 | 74.88 | 0.22 | 0.86 |
| 400 / 1837 | SSLayout360 | 79.67 | 77.09 | 0.25 | 0.93 |
| | SSLayout360* | 79.09 | 76.26 | 0.21 | 0.88 |
| 1650 / 1837 | SSLayout360 | 82.54 | 80.33 | 0.22 | 0.95 |
| | SSLayout360* | 80.89 | 78.13 | 0.19 | 0.90 |