# OpenReview forum: "360-MLC: Multi-view Layout Consistency for Self-training and Hyper-parameter Tuning"
_NeurIPS.cc/2022/Conference — NeurIPS 2022 Accept_

### Official Review · Reviewer_z7v1 · 2022-07-11

**Rating:** 7
**Confidence:** 4
**Soundness:** 3 good
**Presentation:** 4 excellent
**Contribution:** 3 good

**Summary:**

This paper proposes a method to predict room layouts from a set of registered 360 panorama given an arbitrary single-image room layout prediction model trained on a different distribution of data. The main idea is to use reprojected estimates to compute multi-view consistency variance, and then sample from that variance to create pseudo-labels for training. The paper also proposes an entropy metric for evaluating geometric consistency across views. This approach requires no additional ground truth for adapting pre-trained models to new datasets, thus providing an effective way to transfer room layout prediction models to new data.

**Questions:**

* Although this paper presents the multi-view consistency, pseudo-label approach for the task of room layout estimation, the methodology described appears to be more generally applicable. Could the authors share their insight into whether this approach would be transferrable for other tasks in a multi-view setting? For example, object detection, segmentation, or depth estimation.

* One result stands out as curious. While generally the results in Tables 2 and 3 support the proposed method, it is interesting to observe that the entropy metric, $H_{MLC}$ does not particularly favor the approach proposed in the paper. In fact, those tables report similar or even lower entropy (i.e. better consistency) using the simple pre-trained model or the student-teacher approach of [34]. On the other hand, the IoU and depth metrics show consistent significant improvements with the proposed multi-view, self-training approach. Does this indicate that $H_{MLC}$ is simply not an effective metric for evaluating the quality of the model? If so, it would seem to stand counter to the effectiveness of the hyper-parameter tuning shown in Figure 3.

**Limitations:**

This paper does not explicitly address any limitations or social impact of this work.

**Strengths And Weaknesses:**

__Technical Contribution__

This paper presents a reasonably straightforward yet very practical solution for fine tuning a model with multi-view constraints without the need for additional data. The weighted boundary consistency loss in Section 3.2 is an effective way to synthesize the multiple predictions from other views, and the visualizations in Figure 1 and Supplementary Figure 1 provide an intuitive illustration of how it performs. Additionally, the entropy metric seems like an elegant answer to the challenging problem of evaluating results without ground truth data.

This paper provides an in-depth experimental analysis looking at performance improvements on a target dataset when using three different widely-used datasets for training. The quantitative results in Tables 2 and 3 demonstrate clear improvement on standard benchmark metrics compared to the most relevant prior work [34]. Moreover, the paper provides an ablation study that clearly justifies the design decisions of the reprojection loss computation and adds insight to the question of diminishing returns with more images. To the latter question, the results in Table 4(a),(b), and (e) seem to justify the intuition that more reprojection would provide a more stable distribution for pseudo-labels.


__Presentation__

The paper is altogether well written and effectively organized. The writing is clear, and all equations are sufficiently explained. The figures add explanatory value to the paper as well, and the tables are laid out in a clear way that make the conclusions obvious to the reader.

__Summary__

This is a strong paper that provides a simple but seemingly effective method to adapt pre-trained models to new datasets without extensive data labelling. The positive indicators in the quantitative results and the practicality of the solution suggest that this paper will have a reasonable impact on future research into the problem. There is some question about the usefulness of the proposed entropy metric as an evaluation metric in lieu of labels data, but the other hard evaluation metrics show strong gains, justifying the core methodology. This is a somewhat niche application in computer vision--360 room layout estimation--so the paper might benefit from some discussion about the generality of this approach to other problems. However, the relative size of the research community working on this problem should not hinder this paper's acceptance to NeurIPS. I recommend this paper for acceptance.

---

> ### Author Response · Authors · 2022-08-02
> **R4: Response to Reviewer z7v1**
>
> **Q1. Could the authors share their insight into whether this approach would transfer to other tasks in a multi-view setting? For example, object detection, segmentation, or depth estimation.**
>
>  We believe that as long as a set of multi-view recognition results from the same static scene is projected into the same space, the concept of multi-view consistency (MLC) can be used to generate reliable pseudo-labels. For instance, if estimated semantic segments from multi-views can be projected into the same 2D Bird-Eye-View (BEV) or 3D world space, multi-view consistent semantic pseudo-labels can be generated. A similar strategy can also be used for depth estimation and object detection.
>
> Moreover, evaluating the entropy value of multiple estimations from the same modality may also bring important information regarding the overall performance of a model, which comes in handy in the absence of ground truth annotations. This is because a lower entropy is likely correlated with more accurate estimations.
>
> **Q2. Does this indicate that HMLC is simply not an effective metric for evaluating the quality of the model? If so, it would seem to stand counter to the effectiveness of the hyper-parameter tuning shown in Figure 3.**
>
> As the reviewer mentioned, we partially agree that our $\mathcal{H}_\text{MLC}$ metric is not the best metric to measure the performance for layout estimation compared to ground-truth-based metrics such as the 2D/3D IoU. This is because the $\mathcal{H}_\text{MLC}$ metric indirectly measures the layout geometry by projecting all layout boundaries into a 2D top-down grid, as depicted in Figure 2 of our main manuscript. This leads to a measure with a lower resolution due to grid discretization. However, as shown in Figure 3, the $\mathcal{H}_\text{MLC}$ metric can bring important clues about the state of the training, such as the speed of convergence, plateau points, and overall performance. We will clarify this in the revised manuscript.
>
> **Q3. About our limitations.**
>
> The main limitation of our approach is that our multi-view layout consistency (MLC) relies on several views from the same scene with their registered camera poses. Hence, a high-quality sequence of images needs to be recorded to avoid severe illumination changes, drastic camera movements, and textureless regions, which otherwise may lead to wrong camera pose estimations and unreliable pseudo-labels. To verify one solution to our limitation, we replace the ground truth camera pose with estimated camera poses from OpenVSLAM [Sumikura et al., 2019]. The results below show that our proposed method also works well, given a sequence of images with estimated camera poses. For completeness, we will add this limitation to our main manuscript in the revised version.
>
> | Model Description| 2DIoU (%) | 3DIoU (%) | RMSE |$\delta_1$| $\mathcal{H}_\text{MLC}$ |
> |-|-|-|-|-|-|
> | Pre-trained model (source domain MatterportLayout [48]) | 65.38 | 62.28 | 0.58 | 0.78 | 8.18  |
> | 360-MLC with GT poses | 71.50 | 67.20 | 0.49 | 0.78 | 8.14  |
> | 360-MLC w/o GT poses | 70.85 | 66.91 | 0.48 | 0.78 | 8.15  |

---

### Official Review · Reviewer_CFmm · 2022-07-11

**Rating:** 6
**Confidence:** 3
**Soundness:** 3 good
**Presentation:** 4 excellent
**Contribution:** 3 good

**Summary:**

The paper proposes a method to transform a model that estimates room layout boundaries from multiple views from a source to a target dataset. This allows finetuning an existing model on a new dataset without ground truth labels.


**Questions:**

* I am wondering what was the reasoning of only comparing against the SSLayout360 method and not against other methods mentioned in the literature review, such as [4, 22]?

* How does this method compare against approaches that have access to large amounts of training data? Is labeling a lot of data worth the effort or not?

**Limitations:**

I can see that limitations and ethical considerations are not discussed in this paper. Consider adding them.

**Strengths And Weaknesses:**

Strengths:
* The idea to finetune a model based on multi-view boundary consistency is interesting, as it allows finetuning a model without additional labeled data on the target dataset. The proposed method only requires calibrated camera views, and the proposed loss is effective, as shown by the experiments.
* The proposed ablation studies properly motivate the design choices taken for the loss.
* The paper is very well written and easy to follow. The paper also contains a supplementary document that adds missing details for reproducibility and explanations for the used camera projection model.
* The qualitative evaluations show that the model finetuned with the proposed loss cleans up the noisy predictions of the pretrained model.

Weaknesses:
Overall, I do not have many points of criticism with this paper. The paper is well-motivated, the idea is simple yet effective, and the paper has been thoroughly polished.
* In my opinion, the experiments should show a comparison to a model that uses labels from the target dataset as an upper bound of the achievable performance. This will help put the proposed method into perspective.

* Only comparing against SSLayout360 undermines the claim that the method has been extensively tested against other state-of-the-art methods. Consider adding more baselines, such as [4, 22].

---

> ### Author Response · Authors · 2022-08-02
> **R3: Response to Reviewer CFmm**
>
> **Q1. I am wondering what was the reasoning for only comparing against the SSLayout360 method and not against other methods mentioned in the literature review, such as [4, 22]?**
>
> The main motivation for selecting SSLayout360 as the baseline is because this method is the current state-of-the-art semi-supervised method that addresses room layout estimation directly. To the best of our knowledge, a self-training approach for layout geometry estimation has not been proposed yet.
>
> Although both solutions [4] and [22] are representative methods for self-training and semi-supervised learning, applying these solutions to layout geometry estimation is not straightforward and does not exist yet. We refer to [4,22] as related work in this research field that can be considered for future directions.
>
> **Q2. How does this method compare against approaches that have access to large amounts of training data?**
>
> We compare our results using three pre-trained models trained in the largest and most popular datasets for room-layout estimation, i.e., MatterportLayout [48], Zillow (ZInD) [9], Stanford2D3D, and PanoContext (LayoutNet) [47]. The dataset MP3D-FPE [29], used as the target domain, is also one of the largest datasets for 3D reconstruction using 360-images. To the best of our knowledge, the datasets used in this work are the largest datasets that are currently available.
> On the other hand, since our proposed 360-MLC can leverage a multi-view setting, our solution can be applied to even larger datasets bringing the benefits of training data without requiring ground truth annotations.
>
> **Q3. Is labeling a lot of data worth the effort or not?**
>
> Since the current state-of-the-art models for room-layout estimation are supervised methods, a large amount of labeled data will always have a positive impact on their performance. However, obtaining such labels in custom data within a multi-view setting, as in our work, is a cumbersome process. Therefore, it is why we designed the proposed algorithm to leverage data without needing ground truth annotations. Similarly, obtaining upper-bound results requires significant efforts for labeling our target dataset, which is not considered yet in this work.
>
> **Q4. About our limitations**
>
> The main limitation of our approach is that our multi-view layout consistency (MLC) relies on several views from the same scene with their registered camera poses. Hence, a high-quality sequence of images needs to be recorded to avoid severe illumination changes, drastic camera movements, and textureless regions, which otherwise may lead to wrong camera pose estimations and unreliable pseudo-labels. To verify one solution to our limitation, we replace the ground truth camera pose with estimated camera poses from OpenVSLAM [Sumikura et al., 2019]. The results below show that our proposed method also works well, given a sequence of images with estimated camera poses. For completeness, we will add this limitation to our main manuscript in the revised version.
>
> | Model Description| 2DIoU (%) | 3DIoU (%) | RMSE |$\delta_1$| $\mathcal{H}_\text{MLC}$ |
> |-|-|-|-|-|-|
> | Pre-trained model (source domain MatterportLayout [48]) | 65.38 | 62.28 | 0.58 | 0.78 | 8.18  |
> | 360-MLC with GT poses | 71.50 | 67.20 | 0.49 | 0.78 | 8.14  |
> | 360-MLC w/o GT poses | 70.85 | 66.91 | 0.48 | 0.78 | 8.15  |

---

### Official Review · Reviewer_zKQV · 2022-07-12

**Rating:** 5
**Confidence:** 3
**Soundness:** 3 good
**Presentation:** 3 good
**Contribution:** 3 good

**Summary:**

The paper proposes a framework for finetuning monocular indoor layout estimation models to target domain without ground truth labels. The framework consists of a self-supervise loss using output layouts and uncertainty from multiple views, as well as a metric for evaluating the finetuned model and for hyper-parameter tuning. The paper evaluates against baseline methods both qualitatively and quantitatively in the aforementioned settings, and demonstrates the effectiveness of the proposed scheme.

**Questions:**

Please see the Weakness section for questions.

**Strengths And Weaknesses:**

Strengths:

[1] Novelty. The task setting and the solution is novel, in that a purely self-supervised loss and a metric for finetuning are proposed, which is shown as both intuitive and effective.
[2] Evaluation. The evaluation is sound and comprehensive.

Weakness:
[1] Limited improvements quantitively. On the one hand, limited qualitatively results and comparisons are included in the paper, making it difficult to interpret the effectiveness of the proposed other than the benchmarks. Given the two samples in Fig. 4, the improvement upon baseline methods and pretrained model is limited, where the proposed method shows improvements in some regions but misses out in a few others. The quantitative numbers show improvements, but it is difficult to interpret how the improvement in numbers translate to cross-view consistency (e.g. from BEV). Also limited number of baselines are proposed. I wonder if it is possible to look for or adapt more previous methods to this setting for evaluation.

---

> ### Author Response · Authors · 2022-08-02
> **R2: Response to Reviewer zKQV**
>
> **Q1. Qualitative results.**
>
> To address the question about the limited qualitative results, we show our qualitative results in our supplementary material, where three sets of qualitative results are presented in Figures 1 to Figure 3.
>
> In Figure 1, we present a set of 360-images to demonstrate our proposed multi-view layout consistency MLC. In column (a), all layouts in the scene are projected in the image view as yellow lines. In column (b), the estimated pseudo label for each image is presented as magenta lines. In column (c), the associated uncertainty is presented as image maps. In this figure, we can appreciate that our MLC approach can generate plausible pseudo-labels using only multi-view geometry reprojection.
>
> In Figure 2, we present a set of 2D density maps, similar to the depicted in Figure 2 in our main manuscript. In this figure, we can observe how a pre-trained model, shown in column (a), is improved using our method. In columns (b) and (c), the results of SSLayout360 [34] and our proposed 360-MLC are presented, respectively. Note that the yellow silhouette in these density maps represents a large accumulation of layout boundaries, which shows the quality of geometry consistency.
>
> In Figure 3, we compare layout boundaries projected in the 360-image, where the ground truth label, SSLayout360, and our proposed 360-MLC are depicted as cyan, magenta, and yellow lines, respectively. From these results, we show that our estimated layout geometry is more accurate.
>
> **Q2. Baselines.**
>
> In our experiments, we benchmark our proposed 360-MLC against a pretrained model HorizonNet [30], and the current state-of-the-art SSLayout360 [34]. To the best of our knowledge, a self-training method for layout estimation has not been proposed yet.
>
> To address the concern about using other methods that can leverage our proposed 360-MLC framework, we adopt HoHoNet [Sun et al. 2021] to generate pseudo-labels estimated from our multi-view layout consistency. In this experiment, we use a pretrained HoHoNet model trained in MatterportLayout [48] (source domain) to be self-trained in the MP3D-FPE dataset [29] (target domain) without using ground truth labels.
> The results show that our proposed solution can effectively be used for different room-layout models like HoHoNet.
>
> | Model Description| 2DIoU (%) | 3DIoU (%) | RMSE |$\delta_1$| $\mathcal{H}_\text{MLC}$ |
> |-|-|-|-|-|-|
> | Pre-trained (source domain MatterportLayout [48])|66.01|62.36|0.55|0.78|8.20|
> | 360-MLC (HoHoNet [Sun et al. 2021] backbone)| 70.76| 67.70| 0.47 | 0.80 | 8.15 |

---

### Official Review · Reviewer_eCgj · 2022-07-15

**Rating:** 6
**Confidence:** 5
**Soundness:** 3 good
**Presentation:** 3 good
**Contribution:** 3 good

**Summary:**

The paper presents a self-supervised approach for target domain adaption specifically for 360 room layout estimation. The idea is to leverage the consensus of the predicted layout from multiple panorama of the same scene. A weighted boundary consistency loss is proposed to learn from pseudo ground truth with uncertainty. A multi-view consistency metric is proved to be strongly correlated with the room layout metric and thus can be used to tune hyper-parameters even with no access to the ground truth.

**Questions:**

- How is the camera pose on the target dataset obtained? Is it calculated from color images or relies on external sensors?
- In Tab 4, it seems that disable uncertainty loss, some of the metrics get better. I'd like to hear authors' comments on this.
- Is the SSLayout360 in Tab 1 and 2 trained only on pre-trained dataset and never sees images from target domain?


**Limitations:**

- It seems that the 6dof camera poses of panoramas are required to do the projection. Hence, precisely speaking, the method is not fully self-supervised but requires camera pose ground truth. This is usually accessible, easier compared to the ground truth layout, but may also cause error for the layout projection and thus hurts the overall finetuning performance.

- The experiment could be stronger to demonstrate the effectiveness of the method from two aspects: 1) a stronger baseline. It seems SSLayout360 is in general outperforming HorizonNet. It would be convincing to show that this method is able to improve powerful backbones. 2) analyze the domain gap. It would be nice to add some discussions about the gap between datasets. Some datasets are closer to each other thus the adaption may not be a big issue. Also, if the method is able to finetune a pre-trained model on synthetic data, then the value of the approach would be much higher.

**Strengths And Weaknesses:**

- The method is technically sound. The self-supervised learning is an important and effective solution for domain adaptation.
- Evaluation on multiple datasets show positive contribution by using the proposed method.
- Ablation study is provided to show the contribution of individual technical components.

---

> ### Author Response · Authors · 2022-08-02
> **Response to Reviewer eCgj**
>
> **Q1. How is the camera pose on the target dataset obtained? Is it calculated from color images or relies on external sensors?**
>
> In the experiment presented in our paper, we use ground truth camera poses provided by the MP3D-FPE dataset [29]. The motivation for this decision is to avoid issues related to the odometry estimation, camera distortion, etc., and focus on leveraging multi-view layout estimation directly. As the reviewer mentioned, the registration of multiple cameras in a multi-view setting can be accomplished by external sensors, inertial sensors, or Simultaneous Localization and Mapping (SLAM) solutions.
> To complement the experiments presented in our main manuscript, we conduct experiments using estimated camera poses from OpenVSLAM [Sumikura et al., 2019]. The results show that our proposed method can still generate multi-view consistency pesudo-labels using estimated poses, which can be used for self-training a room-layout model without needing ground truth camera poses at all.
>
> | Model Description| 2DIoU (%) | 3DIoU (%) | RMSE |$\delta_1$| $\mathcal{H}_\text{MLC}$ |
> |-|-|-|-|-|-|
> | Pre-trained model (source domain MatterportLayout [48]) | 65.38 | 62.28 | 0.58 | 0.78 | 8.18  |
> | 360-MLC with GT poses | 71.50 | 67.20 | 0.49 | 0.78 | 8.14  |
> | 360-MLC w/o GT poses | 70.85 | 66.91 | 0.48 | 0.78 | 8.15  |
>
> **Q2. In Tab 4, it seems that disabling uncertainty loss, some of the metrics get better. I'd like to hear the authors' comments on this.**
>
> As the reviewer mentioned, the RMSE and $\delta_1$ metrics in Table 4(e) are slightly worse than the model without $\mathcal{L}_\text{WBC}$ in Table 4(d). This is because both metrics indirectly measure the estimated layout by projecting and rendering a depth map, which reflects a lower resolution in the measurement due to pixel discretization. In the case of RMSE, the root-means-squared error is evaluated over all pixels in the depth map. For $\delta_1$, the percentage of the pixels within a 25% error is measured. Therefore, variations in the layout estimation do not significantly change the depth map at the pixel level, and thus these two metrics do not fully represent the quality of layout estimation.
>
> **Q3. Is the SSLayout360 in Tab 1 and 2 trained only on the pre-trained datasets and never sees images from the target domain?**
>
> No, in both experiments depicted in Table 2 and Table 3, SSLayout360* [34] sees images from the target domain but without annotations. Details for these experiments can be found in L248 and L255 of our main manuscript as Setting1 and Setting 2, respectively. In each setting, all the models are trained with the same data usage.
> For the experiments in Table 2 (Setting 1), ground truth annotations are provided only for the images in the source domain, along with unlabeled target-domain images. For Table 3 (Setting 2). We further experiment with a more practical scenario with a source-free setting, i.e., having the source-domain pre-trained model and unlabeled target-domain images.
>
> **Q4. About using another backbone model**
>
> To corroborate our proposed method further, we adopt HoHoNet [Sun et al. 2021] to the proposed 360-MLC framework by replacing HorizonNet with HoHoNet as the backbone model. In this experiment, we implement HoHoNet with a pre-trained model in MatterportLayout [48] from the official implementation, and then we generate pseudo-labels using our proposed method. The target domain used in this experiment is the MP3D-FPE dataset [29]. The results show that our proposed 360-MLC can be applied to other layout estimation models on a new dataset without using ground truth annotations. These results are comparable with the experiments presented in Table 3 (using Setting 2).
>
> | Model Description| 2DIoU (%) | 3DIoU (%) | RMSE |$\delta_1$| $\mathcal{H}_\text{MLC}$ |
> |-|-|-|-|-|-|
> | Pre-trained (source domain MatterportLayout [48])|66.01|62.36|0.55|0.78|8.20|
> | 360-MLC (HoHoNet [Sun et al. 2021] backbone)| 70.76| 67.70| 0.47 | 0.80 | 8.15 |
>
> **Q5. Domain gap**
>
> We investigate the data domain gap by showing the t-SNE plot of the four datasets used in this work. This plot is presented in our updated supplementary material. It is observed that the three source datasets, i.e., MatterportLayout, ZInD, and LayoutNet, have different data distributions with respect to the target dataset, MP3D-FPE [29]. This is because these source datasets consist of real-world images, whereas our target dataset is rendered from a simulator. Our results show that the proposed method can mitigate the domain gap consistently across the three datasets. We will include the usage of synthetic data in our framework as future work.

---

### Meta-Review · Area_Chair_4Jgf · 2022-08-21

**Recommendation:** Accept
**Confidence:** Less certain

**Metareview:**

The reviewers were all supportive of this paper and commended the author’s efforts to clarify their questions with follow up responses and additional experiments. This area chair agrees and recommends acceptance.

The authors are encouraged to take the reviewer comments on board in the final camera ready paper. This includes the promise to discuss the limitations raised by the reviewers (e.g. z7v1 and CFmm). The additional results with a different backbone and non-GT poses can be included in the supplementary material if there is no space in the main paper.

Minor comments:
Suggested text change in abstract:
“This comes in handy in …” -> “This can be valuable in …”


**Award:**

No

---

### Decision · Program_Chairs · 2022-09-14

Accept